# Enhanced Removal of Sb (III) by Hydroxy-Iron/Acid–Base-Modified Sepiolite: Surface Structure and Adsorption Mechanism

**Yu Zou** [1,2], **Bozhi Ren** [1,2,*], **Zhendong He** [1,2] and **Xinping Deng** [3]

1  Hunan Provincial Key Laboratory of Shale Gas Resource Exploitation, Xiangtan 411201, China
2  School of Civil Engineering, Hunan University of Science and Technology, Xiangtan 411201, China
3  402 Geological Prospecting Party, Xiangtan 411201, China
*  Correspondence: bozhiren@126.com

**Abstract:** To improve the removal of antimony (Sb) from contaminated water, sepiolite (Sep) was chosen as the feedstock, modified with an acid–base and a ferric ion to yield a hydroxy-iron/acid–base-modified sepiolite composite (HI/ABsep). The surface structure of the HI/ABsep and the removal effect of the HI/ABsep on Sb (III) were investigated using potassium tartrate of antimony as the source of antimony and HI/ABsep as the adsorbent. The structural features of the HI/ABsep were analyzed by SEM, FTIR, PXRD, BET, and XPS methods. Static adsorption experiments were performed to investigate the effects of adsorption time, temperature, adsorbent dosage, and pH on the Sb (III) adsorbed by HI/ABsep. This demonstrates that sepiolite has a well-developed pore structure and is an excellent scaffold for the formation of hydroxy-iron. HI/ABsep adsorption of Sb (III) showed the best fit to the pseudo-second-order model and the Freundlich model. The maximum saturated adsorption capacity of the HI/ABsep regarding Sb (III) from Langmuir's model is 25.67 mg/g at 298 K. Based on the research results, the HI/ABsep has the advantages of easy synthesis and good adsorption performance and has the potential to become a remediation for wastewater contaminated with the heavy metal Sb (III).

**Keywords:** antimony; hydroxy-iron; sepiolite; adsorption





## 1. Introduction

The heavy metal pollution of water supplies and sewers has become a common problem in the production of food safety and human health [1]. Antimony and its compounds, as a crucial strategic nonrenewable metal, have been extensively used in flame retardants, alloys, ceramics, pigments, and synthetic fibers [2,3]. However, antimony is a widely distributed toxic element. Antimony (Sb) is classified as one of the main pollutants by the US Environmental Protection Agency and the European Environmental Protection Agency due to its proven and potential carcinogenicity, immunotoxicity, genotoxicity, and toxicity to the reproductive system [4–6]. Antimony in water exists primarily in two forms, Sb (V) and Sb (III), among which the toxicity of Sb (III) is approximately ten times higher than that of Sb (V) [7]. China also has the largest reserve of antimony and is the world's largest coal miner. Extensive antimony mining makes antimony pollution more serious in the southwest [8]. Therefore, it is crucial to find a rapid and efficient way to remove antimony.

In the past few decades, standard methods for removing Sb mainly include adsorption [9], ion exchange [10], coagulation [11], and microbial treatment [12]. The adsorption method is still one of the most widely used of these methods due to its advantages: simple operation, low cost, and greenness of the materials. The adsorbents include modified non-metallic minerals, natural or synthetic metal oxides, and carbon materials, etc. The current research results show that iron oxyhydroxide has a good removal effect on antimony in water [13,14]. However, ferric hydroxide in water is often in a loose, amorphous flocculation

state, with the disadvantage of not being easy to disband nor easy to separate from water. These shortcomings increase the cost of removing antimony from iron oxides and limit their practical application. The key approach to solving this problem is to fix iron oxide in coarse, porous granular carriers with a large specific surface area. At present, iron modification is widely applied in clay minerals. Bahabadi et al. [15] compared the adsorption effects of several naturally occurring clay minerals and iron-modified clay minerals on heavy metal adsorption and found that the modification of $Fe^{3+}$ ions can be localized in the channel or that oxygen- or hydrogen–oxygen complexes are formed at the mineral surface to enhance the active sites of the mineral. Xu et al. [16] prepared a composite material of iron oxide and quartz sand from iron oxide-filled quartz sand and used it to adsorb Sb (III) into the water.

The mineral sepiolite (Sep) is a naturally occurring magnesium-rich silicate clay mineral with a standard crystalline chemical formula of $Mg_8Si_{12}O_30(OH)_4(OH2)_4 \cdot 8H_2O$. China's sepiolite resources are mainly distributed in Hunan Province, with proven reserves exceeding 21.4 million tons. Therefore, it is an excellent choice to study iron modification with sepiolite from local materials. In the unique channel of the sepiolite nanostructure, the presence of "zeolite water" and some $Mg^{2+}$ and $Ca^{2+}$ exchangeable ions cause it to have strong adsorption properties and ion exchange capacities, so its potential application in the heavy metal adsorbing field has been widely investigated [17,18]. Sepiolite is cheap, readily available, and has a large surface area. It has great potential in the treatment of heavy metal wastewater. Natural sepiolite, however, has shortcomings, such as low surface acidity, a small channel, and low metal binding constant, so it needs proper modification processing [19].

Hence, this study is mainly conducted from the following three aspects: (I) the modified two-step sepiolite synthesis of antimony adsorption material, and SEM-EDS, FTIR, PXRD, BET, and XPS technology enhanced for analyzing the adsorbing process. (ii) The surface structure and adsorption conditions of HI/ABsep are investigated, including contact time, adsorbent dosage, pH, and initial concentration of Sb (III). (iii) The adsorption behavior and adsorption type of HI/Absep-adsorbed Sb (III) were determined by quasi-primary and quasi-secondary kinetic models, and Langmuir and Freundlich isothermal models.

## 2. Materials and Methods

### 2.1. Preparation of HI/ABsep

Sepiolite purity >80%, procured from Xiangtan Sepiolite Technology Co., Ltd. (Xiangtan, China). All reagents are of analytical grade and procured from Sinopharm Chemical Reagent Co., Ltd. (Shanghai, China).

Sepiolite (10 g) is dispersed in 2.5 mol/L HCl solution (250 mL) at room temperature, acid activation is performed at 250 r/min agitation speed for 24 h, then ammonia water is added, and added dropwise under magnetic stirring in the solution until the pH of the solution reaches 10.0. The supernatant was stirred for 2 h and then removed by centrifugation, followed by several items of deionized water and absolute ethanol. The precipitate is finally collected, dried in a drying oven at a constant temperature of 80 °C, crushed, and sieved to obtain a combined acid–base-modified sepiolite (abbreviated as ABsep). Disperse ABsep (10 g) in 1 mol/L $FeCl_3$ solution (10 mL) and sonicate for 10 min. Then, 1 mol/L NaOH (10 mL) was added, stirred with a magnetic stirrer at 250 r/min for 2 h, washed several times with deionized water, centrifuged, dried at 80 °C, then ground, continued to be cleaned until nearly neutral, and then after drying, passed through a 200-mesh sieve, and, finally, HI/ABsep is prepared.

### 2.2. Characterization Methods

Before the option of Sb (III) in simulated wastewater, sepiolite and HI/ABsep derived from sepiolite were characterized. Fourier-transform infrared spectroscopy (FTIR) (ALPHA, Bruker) was used to analyze the functional groups in the samples and their changes during the reaction. The surface morphology and structural characteristics of the sample were observed by scanning electron microscope (SEM) (JEOL JSM-6700F) with energy spectrum;

X-ray powder diffraction (PXRD) was obtained by X-ray diffractometer (D/Max 2500, Rigaku). The material pore structure was measured by the specific surface area and porosity analyzer (ASAP2020M). The composition and distribution information of chemical elements was obtained by X-ray photoelectron spectroscopy (XPS) (ESCALAB 250Xi, Thermo fisher, Waltham, MA, USA).

*2.3. Batch Adsorption Experiments*

The powdered potassium tartrate of antimony dissolved in diluted hydrochloric acid was prepared for 500 mg/L, mother liquor pH 6, and diluted by mother liquor to obtain simulated wastewater of different antimony concentrations. The separated or modified sepiolite powders were added to the simulated antimony wastewater and the vibrating adsorption at 1500 r/min under different experimental conditions and requirements. This was filtered through a 0.45 μm syringe diaphragm filter. An atomic absorption spectrometer (TAS-990) was used to determine the equilibrium concentration of Sb (III) in the supernatant. The equilibrium adsorption capacity ($q_e$) and adsorption efficiency $\eta$ (%) of Sb (III) were calculated by the following formula:

$$q_e = \frac{(C_0 - C_e) \times V}{C_0} \tag{1}$$

$$\eta = \frac{C_0 - C_e}{C_0} \times 100\% \tag{2}$$

where $C_0$ and $C_e$ are the initial and equilibrium (mg/L) concentrations of Sb (III), respectively, $V$ is the volume of solution (L), and $m$ is the mass of the adsorbent (g).

2.3.1. Adsorption Kinetics Experiment

Sep (1.0 g) and HI/ABsep (1.0 g) were added to the 20 mg/L Sb (III) solution (500 mL). At this time, the initial pH of the solution was not adjusted (in this case, pH = 6). The beaker with the mixture was transferred to a magnetic stirrer and continuously stirred at 150 r/min speed at room temperature. At each sampling time (3–240 min), samples were collected and immediately filtered through a 0.45 μm syringe membrane filter to determine the equilibrium concentration of Sb (III) in the supernatant.

The pseudo-first-order Equation (3) and pseudo-second-order Equation (4) are used for nonlinear fitting. The formula are as follows:

$$q_t = q_e \left( 1 - e^{-K_1 t} \right) \tag{3}$$

$$q_t = \frac{K_2 q_e^2 t}{1 + K_2 q_e t} \tag{4}$$

where $K_1$ is the pseudo-first-order kinetic equation rate constant (min$^{-1}$); $K_2$ is the pseudo-second-order kinetic equation rate constant (g·mg$^{-1}$·min$^{-1}$); $t$ is the adsorption time (min); $q_t$ and we are the Sb (III) adsorption amount (mg·g$^{-1}$) at time $t$ and adsorption equilibrium, respectively.

2.3.2. Effect of Adsorbent Dosage and pH Value

Experimental conditions stipulate a temperature of 35 °C, a volume of solution 50 mL, an initial concentration of Sb (III) 10 mg/L, an initial pH of the system 6, an adsorption time of constant temperature oscillation 240 min, a dosage of HI/ABsep (0.5–3 g/L), respectively, for the effect on adsorbing heavy metal Sb (III).

In the same way, HCl (0.1–1 mol/L) and NaOH (0.1–1 mol/L) were used to adjust the initial pH value of the adsorption system (2–12), the dosage of the adsorbent is 2 g/L, and the other conditions are the same as mentioned above. Multiple adsorption tests were carried out on heavy metal Sb (III), and the effect of pH on the adsorption of Sb (III) at HI/ABsep was obtained.

### 2.3.3. Adsorption Thermodynamics

Sep and HI/ABsep (0.1 g) were added to a series of 50 mL solutions of Sb (III) (10, 20, 50, 80, 100, and 150 mg/L), respectively. The solution was placed in a box with constant-temperature gas bath agitation and stirred at 150 r/min for 240 min at a temperature of 298 K, 308 K, and 318 K, respectively. Sb (III) was measured by filtering the supernatant through a 0.45 μm membrane syringe filter.

The experimental data were analyzed using two adsorption models. Langmuir Equation (5) and Freundlich Equation (6) perform non-linear fits of the adsorption isotherms at different temperatures (289 K, 308 K, and 318 K), respectively.

$$q_e = \frac{K_L Q_m C_e}{1 + K_L C_e} \tag{5}$$

$$q_e = K_F C_e^{1/n} \tag{6}$$

where $q_e$ is the equilibrium adsorption amount (mg/g); $C_e$ is the equilibrium concentration (mg/L); $Q_m$ is the saturation adsorption amount (mg/g); $K_L$ (L·mg$^{-1}$) and $K_F$ [mg·g$^{-1}$· (L·mg$^{-1}$)$^{1/n}$] are Langmuir adsorption constant and Freundlich adsorption constant, respectively; $1/n$ is a constant for adsorption strength, which varies with the heterogeneity of the material.

### 2.3.4. Thermodynamic Parameters

In order to further explain the spontaneous characteristics and energy changes in the preparation of HI/ABsep when adsorbing Sb (III) in the solid–liquid system, the thermodynamic parameters under three different temperatures (standard free energy $\Delta G°$ standard enthalpy $\Delta H°$, standard entropy $\Delta S°$) were calculated by using Equations (7) [20–22], (8), and (9).

$$K_{Eq}° = \frac{K_T \times C_{Adsorbate}°}{r_{Adsorbate}} \tag{7}$$

$$\Delta G° = -RT ln K_{Eq}° \tag{8}$$

$$ln K_{Eq}° = \frac{\Delta S°}{R} - \frac{\Delta H°}{RT} \tag{9}$$

where $K_{Eq}°$ (dimensionless) is the standard thermodynamic equilibrium constant of adsorption based on molar concentrations; $K_T$ is the distribution coefficient obtained from the Langmuir isotherm (L/mol); $C_{Adsorbate}°$ (mol/L) is the standard concentration of adsorbate; and $r_{Adsorbate}$ (dimensionless) is the activity coefficient of the adsorbate.

Using $ln K_{Eq}°$ and 1/T as the plot, the intercept and slope obtained by linear regression analysis can be used to calculate $\Delta S°$ and $\Delta H°$, respectively, and $\Delta G°$ can be directly calculated by Equation (8).

## 3. Results and Discussion

### 3.1. Characterization of Sep and HI/ABsep

#### 3.1.1. N$_2$ Adsorption–Desorption Isotherm and Pore Size Distribution

Before compounding hydroxy iron into sepiolite, the sepiolite needs to be modified with acid and alkali. The N$_2$ adsorption–desorption isotherm at 77 K and pore size distribution of sepiolite, ABsep, and HI/ABsep are shown in Figure 1a,b. Both of these two materials are of type IV adsorption. Their pores are concentrated in size distribution at 2–50 nm, indicating that there are a large number of mesopores in the sample, and antimony can penetrate the internal sepiolite surface, which is beneficial for adsorption [23]. The mean pore diameters of the ABsep composite and Sep materials calculated by BJH are 1.4 nm and 3.7 nm, respectively. The BET formula gives a significantly higher average specific surface area for the ABsep composite material (260.5 m$^2$/g) than Sep (160.1 m$^2$/g); it can be seen that the surface performance after the acid–base modification has been greatly

improved compared with that before modification, which provides a good foundation for the composite of hydroxyl iron. HI/ABsep surface area and average pore diameter were 152.4 m$^2$/g and 1.7 nm, respectively, both smaller than the surface area and average pore diameter of ABsep. The reason for this may be that, during the introduction of iron and supported iron oxide, some channels are blocked, forming a thick membrane on the surface that prevents N$_2$ from adsorption and desorption [24].

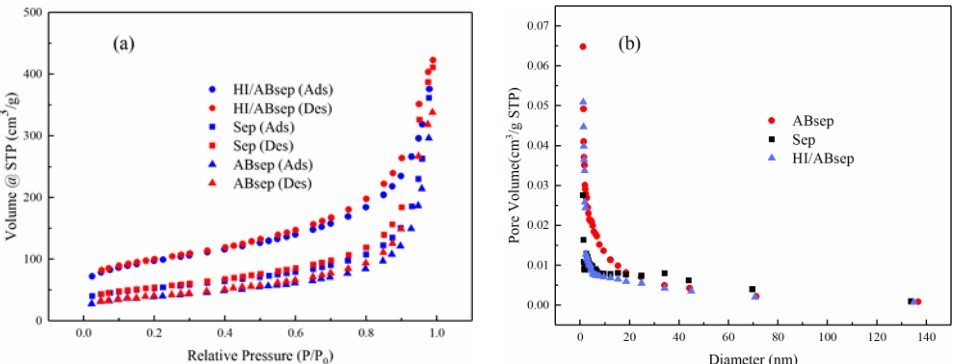

**Figure 1.** N$_2$ adsorption–desorption isotherm at 77 K (**a**) and BJH pore size distribution (**b**) of Sep, ABsep, and HI/ABsep.

### 3.1.2. SEM Analysis

As shown in Figure 2a, the original structure of the sepiolite material is a thin straight rod-like fiber structure. In Figure 2b, the SEM image of the ABsep material obtained by treatment of sepiolite with hydrochloric acid and ammonia is shown. As can be seen, the surface of the composite is non-uniform, and the nanofibers are fragmented in comparison to the SEM image of the original sepiolite structure. With the massive structure, the dispersibility is reduced, the agglomeration appears, and the pores increase. The result is consistent with Figure 1. When compared to the surface without iron modification, the iron-modified HI/ABsep showed irregular ripples and a highly roughened surface (Figure 2c), which could be affected by hydroxyl iron.

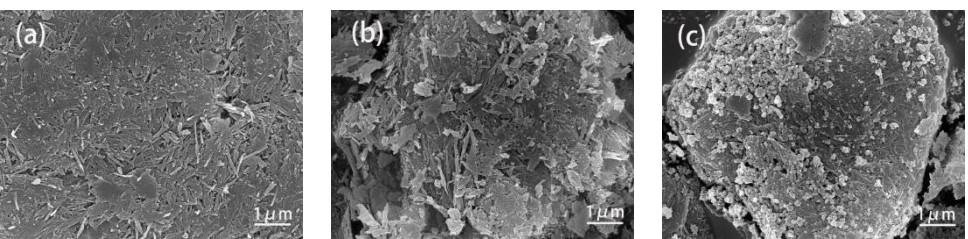

**Figure 2.** SEM image of Sep (**a**) and ABsep (**b**) and SEM-EDS image of HI/ABsep (**c**).

### 3.1.3. PXRD and FTIR Analysis

The PXRD spectrum of the sample is shown in Figure 3b. The Sep spectrum showed apparent diffraction of 7.2, 11.8, 19.7, 23.7, 26.7, 33.5, and 40.2 degrees. The normalized card number ((JCPDS: 14-0001) indicates that (110), (130), (300), (340), (270), (510), and (490) are known to correspond to the sepiolite orders. After modification, the sepiolite peak disappears, and the characteristic diffraction peaks 2θ of HI/ABsep are 20.9°, 26.6°, and 50.1°, which are following SiO$_2$ (JCPDS:46-1045) [25], indicating that the main component of HI/ABsep used in this study is SiO$_2$ [26,27]. In addition, the characteristic peaks of HI/ABsep, (130), (021), (221), and (151), correspond to the angles 2θ = 33.2°, 34.7°, 53.2°, and 59.0°, which are consistent with the standard card number (JCPDS: 81-0464). It shows that the main form of iron supported by modified sepiolite is iron oxyhydroxide FeOOH.

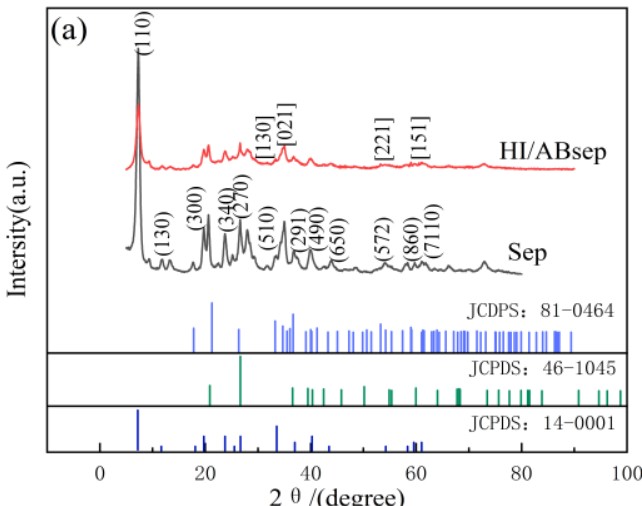
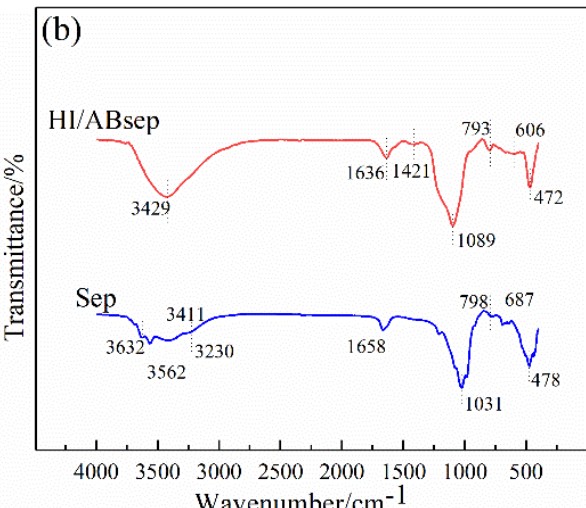

**Figure 3.** PXRD (**a**) and FTIR spectrum (**b**) of Sep and HI/ABsep. The red line in Figure 3a represents the XRD image of HI/ABsep, the black line is the XRD image of Sep, and the light blue refers to the data image corresponding to JCPDS:81-0464; Green is the data image corresponding to JCPDS:46-1045; The dark blue is the data image corresponding to JCPDS:14-0001. The red line in Figure 3b is the image produced by the infrared spectral data of HI/ABsep, and the black line is the infrared spectrum of Sep.

Figure 3a shows the FTIR spectra obtained by Sep and HI/ABsep. In the wavenumber range of 1300–900 cm$^{-1}$, it is a broad and strong stretching vibration band produced by Si-O-Si [28]; the strong absorption band near 3462 cm$^{-1}$ belongs to the structure of the sepiolite ore fine octahedral coordination. The stretching vibration of hydroxyl water (Mg-OH) and coordination water (-OH$_2$), and the absorption peak near 687 cm$^{-1}$ are lattice vibrations of MgO, Al$_2$O$_3$, and Ca$_2$O$_3$ oxides; the band of 479 cm$^{-1}$ corresponds to vibrations on the Si-O-Si plane [29]; the 1658 cm$^{-1}$ belongs to the peak of C=O stretching vibration; moreover, there are some differences in the peaks of the HI/ABsep spectrum. The hydroxyl peak (-OH) at 3429 cm$^{-1}$ is enhanced. The -OH could be the hydroxyl peak of water. The absorption band characteristic at 1421 cm$^{-1}$ can be caused by Cl-vibration. Especially at 900–450 cm$^{-1}$, which is the characteristic absorption peak of metal oxide, the change before and after modification is related to the appearance of FeOOH oxyhydroxide [30,31].

### 3.2. Adsorption of Sep and HI/ABsep on Sb (III)

#### 3.2.1. Adsorption Kinetics

However, as shown in Figure 4, the amount of Sb (III) adsorbed by Sep and HI/ABsep increased rapidly and then stabilized. At 30 min contact time, the rate of Sb (III) removal by HI/ABsep had reached 67.28%. Within 30–120 min, the absorption rate of the material gradually decreases and tends to equilibrium, reaching an adsorption equilibrium at about 180 min. Sb (III) has a maximum adsorption capacity of 6.14 mg/g and a removal rate of 92%%, which is significantly higher than the original sepiolite (2.65 mg/g); the reason is that hydroxy-iron increases the number of active adsorption sites on the sepiolite surface and the adsorptive capacity is enhanced. The initial rapid adsorption may be due to the external diffusion of Sb (III), which is attached to the external surface of the material. Its main functions are ion exchange and physical adsorption. However, as time increases, on the one hand, the surface of the adsorbent can be used for the adsorption of Sb (III). However, as the concentration gradient of Sb (III) on the solution surface and adsorbent is reduced, the driving force for the concentration gradient decreases so that the resistance of the Sb (III) to adsorbate in solution increases, and the rate of adsorption slowly decreases [32,33].

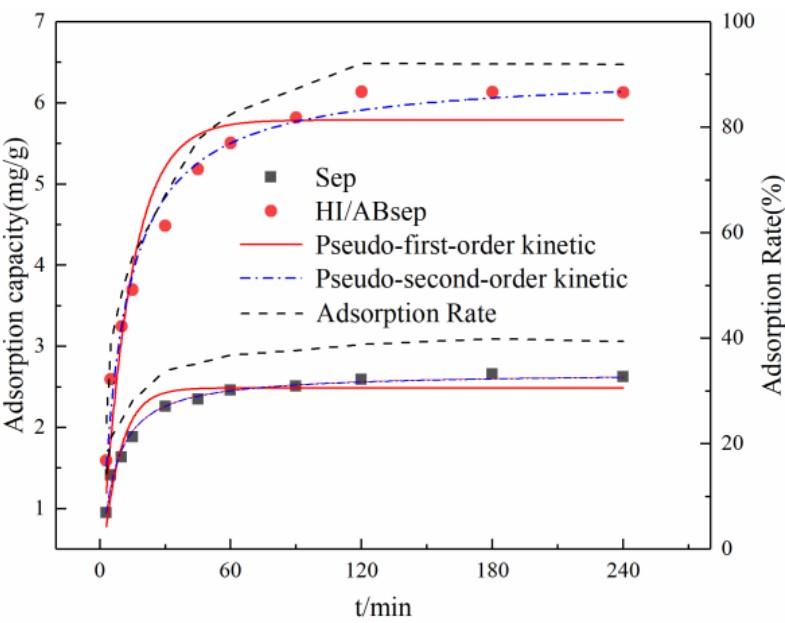

**Figure 4.** Adsorption kinetics of Sep and HI/ABsep.

In Table 1, the $R^2$ value of the quasi-second-order kinetics for the adsorption of Sb (III) by HI/ABsep is 0.9865, which is greater than that of the fitting quasi-first-order kinetics, suggesting that the quasi-second-order kinetics can more accurately describe the adsorption process of Sb (III) by HI/ABeep. Based on the assumption of the quasi-second-order kinetic model, combined with the related literature, the HI/ABsep adsorption process of Sb (III) can be known to be primarily chemical adsorption. The adsorbent and Sb (III) may form chemical bonds by sharing or exchanging electrons. Adsorption occurs through complexation, coordination, or ion exchange [6,9,13,34–37].

**Table 1.** Kinetic parameters of adsorption on Sb (III).

| Adsorbent | Pseudo-First-Order Kinetic Parameters | | | Pseudo-Second-Order Kinetic Parameters | | |
|---|---|---|---|---|---|---|
| | $q_e$ (mg/g) | $K_1$ (min$^{-1}$) | $R^2$ | $q_e$ (mg/g) | $K_2$ (min$^{-1}$) | $R^2$ |
| Sep | 2.4875 | 0.1246 | 0.9078 | 2.6803 | 0.0672 | 0.9865 |
| HI/ABsep | 5.7879 | 0.0769 | 0.9158 | 6.3861 | 0.0163 | 0.9825 |

### 3.2.2. Effect of HI/ABsep Dosage and pH on the Removal of Sb (III)

It has obvious practical significance for the industrial application of sorbent to study the effect of the amount of sorbent and pH on sorbent adsorption performance and the maximum adsorption effect of the sorbent on different heavy metal ions. From Figure 5a, it can be seen that, as the dosage of HI/ABsep is increased, the rate of elimination gradually increases. When the dosage is 2 g/L, the removal rate reaches 95.46%. Further analysis shows that, when the dosage is low, the number of adsorption sites that can be provided is correspondingly more negligible. The adsorption site of the adsorbent material has a positive correlation with the adsorption efficiency. As a result, the rate of Sb (III) elimination by HI/ABsep at low doses is lower. When the dosage of the adsorbent gradually increases, on the one hand, the adsorption site of the solution also increases rapidly. Conversely, the probability that the reaction between the adsorbed substance and the adsorbate collide with each other also increases as a result; thus, the removal rate of Sb (III) is rapidly increased. Taking into account the economic benefits, we use a dose of 2 g/L in other single-factor experiments.

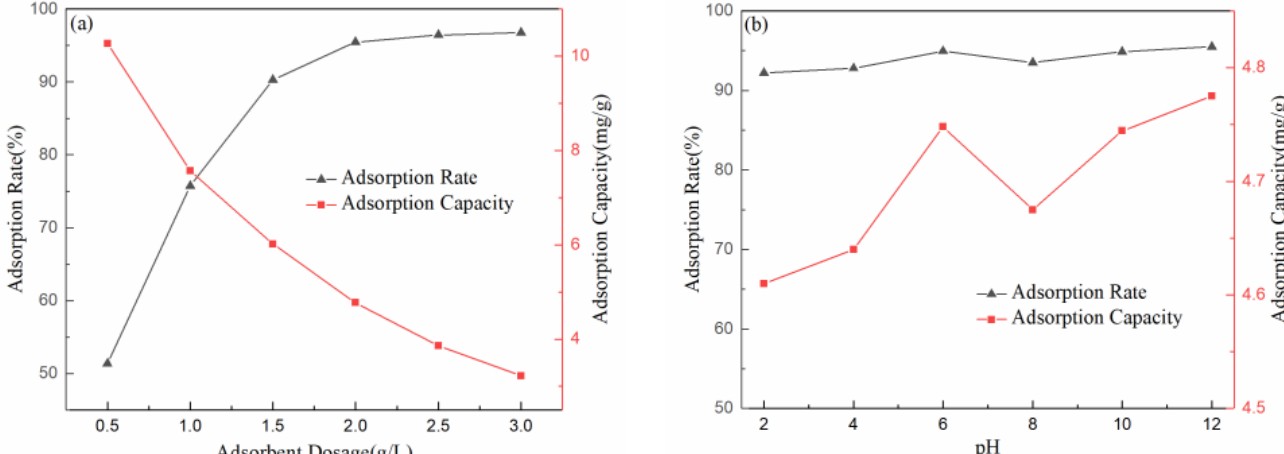

**Figure 5.** Effect of different dosages of adsorbent and pH change on Sb (III) adsorption. (**a**). Effect of different doses of adsorbents on the adsorption of Sb (III); (**b**). Effect of pH on the adsorption of Sb (III) by adsorbents.

From Figure 5b, it can be seen that the pH of the solution does not significantly affect the adsorption efficiency of Sb (III) by HI/ABsep. When pH is 2–6 and pH = 8–12, the removal rate of Sb (III) increased slowly. When pH is 6–8, the removal rate of Sb (III) decreased slowly. On the whole, the removal rate of Sb (III) remained at a high level (92–95.5%). When the pH of the solution is 2~10, the main form of Sb (III) in the solution is the neutral complex Sb (OH)$_3$. Under the strong acid condition, Sb (III) mainly exists as SbO$^+$; all of them may be adsorbed on the surface of goethite with a positive charge as inner sphere complexes [38]. Under the strong alkali condition, its preferential form is SbO$_2$ [35,39–41]. Sb (OH)$_3$ easily reacts with Fe ions to form X $\equiv$ Fe-Sb (OH)$_2$ precipitate [41], and the reaction process is little affected by pH, so the pH has little effect on the adsorption and removal of Sb (III) by HI/ABsep.

### 3.2.3. Adsorption Isotherm

As can be seen from the nonlinear fitting results in Figure 6 and related parameters in Table 2, the Freundlich model has a higher temperature than the Langmuir model ($R^2$ = 0.9595, 0.9062 and 0.9545) at 298 K, 308 K, and 318 K temperatures. The values of the correlation coefficient are 0.9671, 0.9854, and 0.9660. This result demonstrates that the Freundlich model can describe the process of Sb (III) adsorption onto HI/ABsep well. Furthermore, at different temperatures, the Langmuir and Freundlich isotherm models can fit well. This shows that the intermolecular forces play a major role in the magnitude of the intermolecular forces: The higher the temperature, (1) the stronger the adsorption and (2) The greater the concentration of the solution. Langmuir isotherms are appropriate for single-layer adsorption where the adsorption sites are uniformly distributed over the adsorbent surface and there is no interaction between adjacent sites and the particles [42]. The Freundlich adsorption isotherm model is an empirical formula for multi-layer adsorption [43]. It can be deduced that Sb (III) is dominated by the uniform adsorption of multiple molecular layers under a variety of temperature conditions. Furthermore, the range of values of 1/n of the adsorption models of the Langmuir isotherm and Freundlich isotherm is from 0 to 1, indicating that the prepared HI/ABsep has a high affinity to Sb (III), and adsorption may spontaneously develop in the direction of adsorption [44]. According to the Langmuir isotherm (HI/ABsep) adsorption model, the maximum adsorption capacity of Sb (III) is found to be 25.73 mg/ at 298 K.

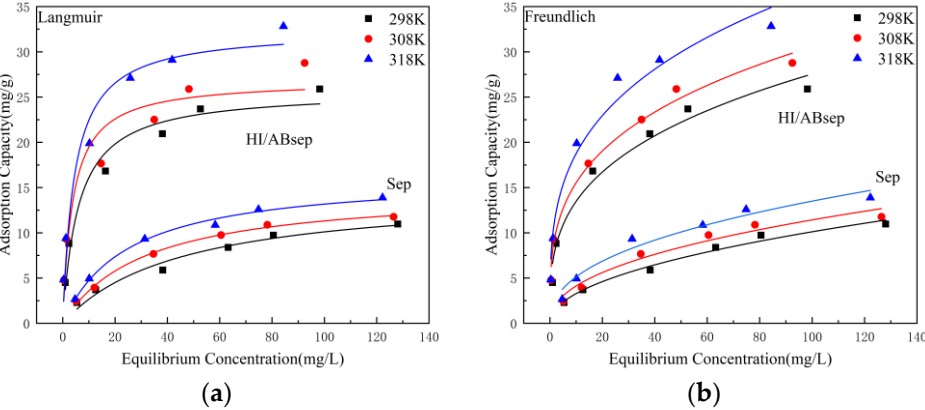

**Figure 6.** Isothermal adsorption line of HI/ABsep to Sb (III). (**a**). the Langermuir isotherm model for HI/ABsep adsorbed Sb (III), 298K, 308K, 318K; (**b**). the Freundlich isotherm model for HI/ABsep adsorbed Sb (III), 298K, 308K, 318K.

**Table 2.** Two adsorption isotherm-related parameters.

| Adsorbent | T/K | Langmuir | | | Freundlich | | |
|---|---|---|---|---|---|---|---|
| | | $K_L$/L·mg$^{-1}$ | $Q_m$/mg·g$^{-1}$ | $R^2$ | $K_F$/(L·mg$^{-1}$)$^{1/n}$ | $n$ | $R^2$ |
| Sep | 298 | 0.0222 | 14.66 | 0.9654 | 1.0643 | 2.0454 | 0.9807 |
| | 308 | 0.0318 | 14.87 | 0.9952 | 1.4890 | 2.2601 | 0.9560 |
| | 318 | 0.0406 | 16.42 | 0.9922 | 1.9511 | 2.3808 | 0.9573 |
| HI/ABsep | 298 | 0.1673 | 25.73 | 0.9595 | 6.5908 | 3.2202 | 0.9671 |
| | 308 | 0.2617 | 26.89 | 0.9062 | 7.8975 | 3.4044 | 0.9854 |
| | 318 | 0.2245 | 32.45 | 0.9545 | 9.5873 | 3.4328 | 0.9660 |

The actual maximum adsorption capacity of the adsorbent HI/ABsep prepared in this study under optimal experimental conditions is 25.73 mg/g, which has adsorption potential compared with other Sb (III) adsorption materials, as shown in Table 3 below. At the same time, considering that the adsorption material in this study has the advantages of simple preparation process and easy physical separation from the treatment system, it has great application potential for the treatment of Sb (III)-contaminated wastewater

**Table 3.** Comparison of Sb (III) adsorption by various adsorbents.

| Sorbents | $Q_m$ (mg/g) | Experimental Conditions | Reference |
|---|---|---|---|
| PVA-Fe$^0$ | 6.99 | pH = 7; T = 298 K | [45] |
| $\alpha$-Fe$_2$O$_3$ | 23.23 | pH = 4; T = 298 K | [14] |
| $\gamma$-FeOOH | 33.08 | pH = 4; T = 298 K | [14] |
| Graphene | 10.919 | pH = 11; T = 298 K | [37] |
| $\gamma$-Fe$_2$O$_3$ mesoporousmicrospheres | 47.48 | pH = 7; T = 298 K | [46] |
| HI/ABsep | 25.73 | pH = 6; T = 298 K | This work |

### 3.2.4. Thermodynamic Parameters

As can be seen from Table 4, as the temperature is increased from 298 K to 318 K, the adsorption of Sb (III) on HI/ABsep has $\Delta G° < 0$ and $\Delta H° > 0$, indicating that the adsorption is a spontaneous endothermic reaction. If the absolute value of $\Delta H$ is between 0 and 20 kJ/mol, the adsorption reaction is related to physical adsorption, and if the absolute value of $\Delta H$ is between 40 and 80 kJ/mol, then the adsorption reaction is related to chemical adsorption [47]. The value of $\Delta H°$ used in this study is 160.18 kJ/mol at 298–318 K, which indicates that the adsorption of Sb (III) by HI/ABsep is chemical adsorption [48]. The

mechanism of adsorption may be an electrostatic force and the filling of pores. This binding increases with an increasing temperature, indicating that increasing temperature is beneficial for the progress of adsorption.

**Table 4.** Thermodynamic parameters for Sb (III) adsorption by HI/ABsep.

| Temperature (K) | $\Delta G°$ (kJ/mol) | $\Delta H°$ (kJ/mol) | $\Delta S°$ (J/mol × K) |
|---|---|---|---|
| 298 | −1.3062 | | |
| 308 | −6.4172 | 160.18 | 0.5409 |
| 318 | −11.8262 | | |

### 3.2.5. The Mechanism of HI/ABsep Adsorption on Sb (III)

In order to gain insight into the HI/ABsep adsorption mechanism of Sb (III), XPS is used to analyze the existence of elements on HI/ABsep and their interaction during the process of adsorption. The high-resolution spectra of HI/ABsep O and Sb before and after adsorption are shown in Figure 7a. The characteristic peaks of 531.84 eV binding energy and 532.53 eV binding energy in the pre-adsorption O1s high-resolution spectrum correspond mainly to Fe-O-OH and Si=O [49] (SiO$_2$), respectively. After adsorption, the two characteristic peaks at the binding energy of 530.65 eV and the binding energy of 540.28 eV correspond to the positions of the standard binding energy peaks of the two orbitals of Sb 3d$_{5/2}$ and Sb 3d$_{3/2}$, respectively. Among the valence states are Sb (III) and Sb (V), which indicates that the redox reaction took place during the adsorption process [50]. The binding energies for Fe 2p3/2 and Fe 2p1/2 of HI/ABsep are about 710.96 eV and 724.26 eV (Figure 7b). The photoelectron peak at 710.96 eV, which is characteristic of iron oxide species in HI/ABsep, shifts to the position of 710.72 eV in HI/ABsep-Sb, which indicates the interaction between adsorbed Sb (III) and iron oxide in HI/ABsep [51]. The removal mechanism of HI/ABsep-strengthened removal of Sb (III) is shown in Figure 8.

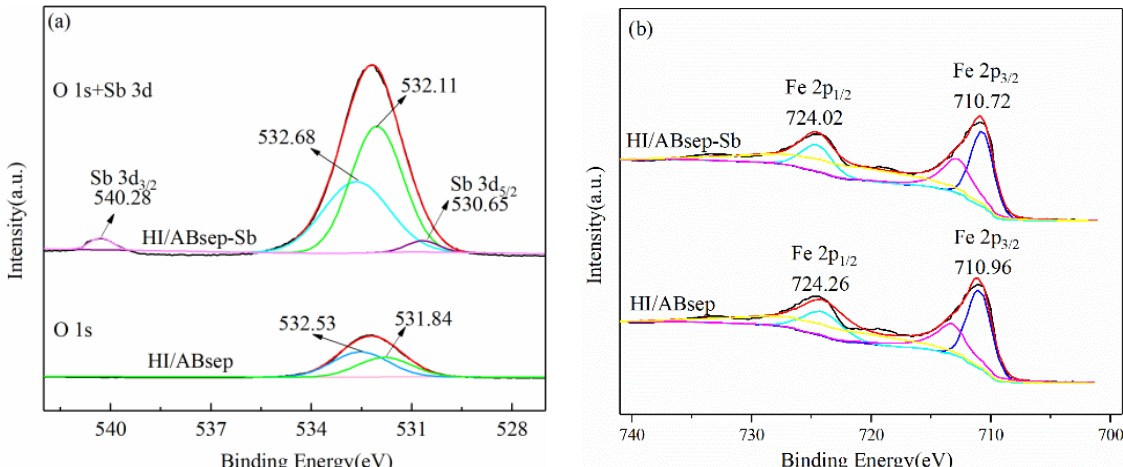

**Figure 7.** XPS spectra before and after HI/ABsep adsorption on Sb (III). (**a**) is the fine spectrum of O1s before and after the reaction, and (**b**) is the fine spectrum of Fe2p before and after the reaction). The red line in Figure 7a indicates the peak fitted to the data, indicating that it is composed of the peaks in the green, cyan, and purple lines; the green and cyan lines represent the two characteristic peaks of O1S corresponding to FeOOH and Si=O; and the purple line represents the characteristic peak of Sb (III); The XPS spectra before and after the adsorption of Sb (III) by HI/ABsep are shown in Figure 7b, respectively. The red line indicates the peak made by fitting the data, and the fuchsia line, cyan line and blue line are made by splitting the peak from the red line. The fuchsia line and the blue line indicate the two satellite peaks of the Fe2p3/2 orbit. The cyan line indicates the characteristic peak of Fe2p1/2 orbit.

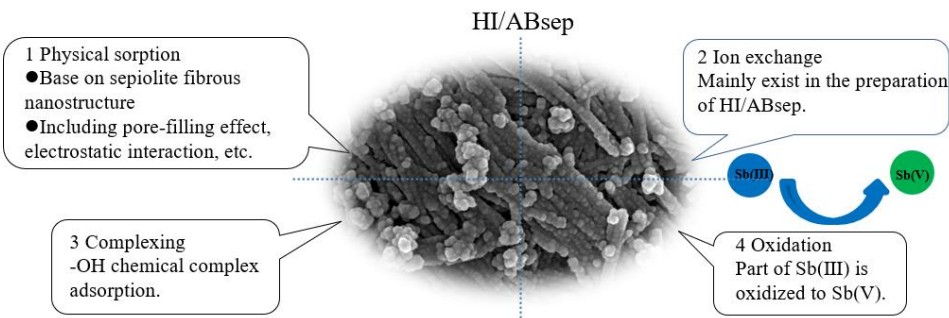

**Figure 8.** Mechanism of Sb (III) Removal by HI/ABsep.

Briefly, the adsorption of HI/ABsep Sb (III) is mainly chemical and physical adsorption. HI/ABsep adsorption onto Sb (III) is a very complex process, which can be combined through a variety of adsorption mechanisms such as redox, coordination complexation, and physical adsorption. A schematic diagram of hydroxyl iron formation and the mechanism of enhanced Sb (III) removal by HI/ABsep can be seen in Figure 8. At pH = 2–10, the main forms of antimony in water are $Sb (OH)_3$ and $Sb (OH)_6^-$. It is speculated that the surface complexation between antimony ions in the simulated water and the hydroxyl iron on the surface of HI/ABsep is shown in the following formula:

$$\equiv MOH + Sb(OH)_3 = MOSbO_2 + 2H_2O \tag{10}$$

$$2 \equiv MOH + Sb(OH)_3 = MOSb(OH)OM + 2H_2O \tag{11}$$

$$Sb(OH)_3 + 3H_2O - 2e^- = Sb(OH)_6^- + 3H^+ \tag{12}$$

$$\equiv MOH + Sb(OH)_6^- = MOSb(OH)_5^- + H_2O \tag{13}$$

$$2 \equiv MOH + Sb(OH)_6^- = MOSb(OH)_4^- OM + 2H_2O \tag{14}$$

where *M* stands for iron, silicon, magnesium, etc.

## 4. Conclusions

HI/ABsep modified based on sepiolite can be used as a promising material for adsorbing heavy metal Sb (III). XRD and XPS show that the Fe species supported on sepiolite are mainly FeOOH. Furthermore, the dispersion of FeOOH on sepiolite led to the exposure of a few new adsorption locations. The kinetics of adsorption follow a pseudo-second-order kinetic model. The Langmuir and Freundlich isotherm models were used to fit the composite adsorbent kinetics successfully, and the maximum adsorption capacity is 25.73 mg/g at 298 K, 26.89 mg/g at 308 K, and 32.45 mg/g at 318 K.

HI/ABsep adsorption of Sb (III) is barely affected by pH but is proportional to the content of the adsorbent. Future research will focus on the practical implementation of the live demonstration.

**Author Contributions:** Conceptualization, B.R. and Y.Z.; Methodology, Y.Z. and Z.H.; Software, Z.H.; validation, Y.Z., Z.H. and X.D.; formal analysis, Y.Z.; investigation, X.D.; Resources, X.D.; Data curation, Y.Z. and Z.H.; Writing—original draft, Y.Z. and Z.H.; Writing—review & editing, B.R. and Y.Z. All authors have read and agreed to the published version of the manuscript.

**Funding:** This work was financially supported by the National Natural Science Foundation of China (Nos. 41973078) and the Hunan Provincial Natural Science Foundation of China (2022SK2073).

**Institutional Review Board Statement:** The study did not deal with ethics.

**Informed Consent Statement:** The study did not involve humans.

**Data Availability Statement:** The study did not report any data.

**Acknowledgments:** We sincerely thank the National Natural Science Foundation of China and the Hunan Provincial Natural Science Foundation of China for their support in experiments and data analysis.

**Conflicts of Interest:** The authors declare no conflict of interest.

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
