# Peer review of "Enhanced Removal of Sb (III) by Hydroxy-Iron/Acid–Base-Modified Sepiolite: Surface Structure and Adsorption Mechanism"

_water, doi:10.3390/w14233806_

Round 1
Reviewer 1 Report
1. Line 186-187, SiO2 need be changed.
2. Line 199, 3429-1 need be changed.
3. Figure 3. PXRD (a) and FTIR spectrum (b) of Sep and HI/ABsep. It should marks the standard card. The -OH could be the hydroxyl peak of water.
4. Line 228, R2 value need be changed.
5. Adsorption occurs through complexation, coordination, or ion exchange. It should be provide more proof, such as Journal of Hazardous Materials 442 (2023) 130024, Chinese Chemical Letters 31(10), 2525-2538.
6. Line 240, Sb(III)Removal need be changed.
7. Figure 5: pH change on Sb(III) adsorption. pH 11 and 12 maybe not good.
8. Line 262. Freund, Line 270. Tab.2, Line 284. Longmuir, Freud Line 292. Tab.3, need be changd. The mechanism can also discussed Trans. Nonferrous Met. Soc. China 28(2018) 980-988.
9. The maximum adsorption capacity of Sb (III) was found to be 25.73 mg/ at 25°C. It should be change the tense.
10. Table 3. Comparison of Sb(III) adsorption by various adsorbents. It should be checked again.
11. Line 321. was, Line 326. Si\\O, need be changed.
12. Ref 35, 38, 21, Such as CHEM ENG J 247(-):250-257. γ-Fe2O3.
Author Response
Thanks for your review, I have made changes to the article. Please see the attachment.

Reviewer 2 Report
After reading your manuscript, I have the following questions and comments:
1. All formulas must be in the Methods section.
2. Formulas 1 and 2 need clarification. Why is there 1000 in formula 1 and 100 in formula 2?
3. Figures 1, 3, 4, 6 need clarification. Notice where the 0 is on the X and Y axis.
4. The literature is not designed according to the requirements of the MDPI.
5. In the Methods section, it is not written how the authors calculated R2.
6. The authors use adsorption models, but the Introduction does not analyse the possibility of using such models.
7. Conclusions contain incorrect theses. Langmuir and Freundlich models are used to determine the type of adsorption (monolayer or not). These models are used to determine the maximum adsorption capacity of the adsorbent. The Authors did not write this. After comparing the value of R2, which was obtained by modelling and using models L and F, the scientist should say which model best describes the adsorption process.
Only the range of adsorption capacity values is written in the conclusions. It is necessary to write that at a temperature of 25 C, the maximum adsorption capacity is ...., at a temperature of 45 C, the maximum adsorption capacity is -....
8. The manuscript must use the same units of measurement of temperature! The authors use K and C!!!
In my opinion, the manuscript is methodically written incorrectly and needs careful rewriting.
With best regards
Author Response
Thanks for your review, we have made correction according for your suggestions. Please see the attachment.

Reviewer 3 Report
The review topic is really interesting and the manuscript is well-written. Therefore, the manuscript has some problems that are listed below:
1) What is the novelty of your study? It is not clear in the Introduction section.
2) What is the common Sb concentration found in the environment? Why did you use 20 mg/L Sb(III)?
3) Please, improve the discussion about the pH effect on the adsorption. Do you have the charges involved in the process? It is necessary to discuss this part of your results.
4) Please, signalize the Tables (2-4) in the manuscript.
5) The discussion of the results with those found in the literature (Table 3) is poor. Please, improve it.
6) Please, signalize the Tables (2-4) in the manuscript.
7) Figure 7 is not clear which figure is (a) and (b). Please, rewrite the caption.
8) Please, correlate the XPS spectra with the FTIR spectra, to indicate the functional group involved in the adsorption process.
Author Response
Thanks for your review, we have made correction according to your suggestions. Please see the attachment.

Round 2
Reviewer 2 Report
I agree such vertion of manuscript. Please, correct literatury as in MDPI